# Endotoxin Inflammatory Action on Cells by Dysregulated-Immunological-Barrier-Linked ROS-Apoptosis Mechanisms in Gut–Liver Axis

**DOI:** 10.3390/ijms25052472

**Published:** 2024-02-20

**Authors:** Andrei Dumitru, Elena Matei, Georgeta Camelia Cozaru, Anca Chisoi, Luana Alexandrescu, Răzvan Cătălin Popescu, Mihaela Pundiche Butcaru, Eugen Dumitru, Sorin Rugină, Cristina Tocia

**Affiliations:** 1Gastroenterology Department, “Sf. Apostol Andrei” Emergency County Hospital, 145 Tomis Blvd., 900591 Constanta, Romania; dr.andreidumitru@gmail.com (A.D.); alexandrescu_l@yahoo.com (L.A.); eugen.dumitru@yahoo.com (E.D.); cristina.tocia@yahoo.com (C.T.); 2Medicine Faculty, “Ovidius” University of Constanta, 1 Universitatii Street, 900470 Constanta, Romania; razvan.popescu@365.univ-ovidius.ro (R.C.P.); butcaru.pundiche@365.univ-ovidius.ro (M.P.B.); sorinrugina@yahoo.com (S.R.); 3Center for Research and Development of the Morphological and Genetic Studies of Malignant Pathology, “Ovidius” University of Constanta, 145 Tomis Blvd., 900591 Constanta, Romania; georgiana.cozaru@365.univ-ovidius.ro (G.C.C.); anca.chisoi@365.univ-ovidius.ro (A.C.); 4Clinical Service of Pathology, “Sf. Apostol Andrei” Emergency County Hospital, 145 Tomis Blvd., 900591 Constanta, Romania; 5Medical Sciences Academy, 1 I.C. Bratianu Street, 030167 Bucharest, Romania; 6Academy of Romanian Scientist, 3 Ilfov Street, 050044 Bucharest, Romania

**Keywords:** gut–liver axis, oxidative stress, cell apoptosis, endotoxins, pro-inflammatory biomarkers, intestinal dysbiosis, acute or chronic intestinal inflammation, liver cirrhosis, hepatic steatosis, dysregulated immune gut barrier

## Abstract

Our study highlighted the immune changes by pro-inflammatory biomarkers in the gut–liver-axis-linked ROS-cell death mechanisms in chronic and acute inflammations when gut cells are exposed to endotoxins in patients with hepatic cirrhosis or steatosis. In duodenal tissue samples, gut immune barrier dysfunction was analyzed by pro-inflammatory biomarker expressions, oxidative stress, and cell death by flow cytometry methods. A significant innate and adaptative immune system reaction was observed as result of persistent endotoxin action in gut cells in chronic inflammation tissue samples recovered from hepatic cirrhosis with the A-B child stage. Instead, in patients with C child stage of HC, the endotoxin tolerance was installed in cells, characterized by T lymphocyte silent activation and increased Th1 cytokines expression. Interesting mechanisms of ROS-cell death were observed in chronic and acute inflammation samples when gut cells were exposed to endotoxins and immune changes in the gut–liver axis. Late apoptosis represents the chronic response to injury induction by the gut immune barrier dysfunction, oxidative stress, and liver-dysregulated barrier. Meanwhile, necrosis represents an acute and severe reply to endotoxin action on gut cells when the immune system reacts to pro-inflammatory Th1 and Th2 cytokines releasing, offering protection against PAMPs/DAMPs by monocytes and T lymphocyte activation. Flow cytometric analysis of pro-inflammatory biomarkers linked to oxidative stress-cell death mechanisms shown in our study recommends laboratory techniques in diagnostic fields.

## 1. Introduction

In healthy conditions, the gut–liver axis assures organism-microbiota homeostasis [1,2]. The gut barrier plays a vital role in host defense by the activating of enterocytes, goblet cells, Paneth cells, and immune cells [3,4]. A group of sentinel goblet cells activates the NLRP6 inflammasome by ROS synthesis, and macrophages from intestinal lamina propria vasculature ensure defense against pathogens [5]. The liver contains parenchymal cells and non-parenchymal cells, including liver sinusoidal endothelial cells, hepatic stellate cells, Kupffer cells, B cells, and T cells. Due to anatomical location and structure, LSEC is the first in contact with portal-delivered gut-derived pathogens, representing the hepatic barrier [6].

Diseases related to chronic alcohol consumption determine liver inflammation, leading to hepatic cirrhosis and hepato-carcinoma [7]. Kupffer cells secrete pro-inflammatory cytokines and reactive oxygen species (ROS) in the liver, determining hepatic stellate and endothelial cell activation [8,9,10]. Non-alcoholic fatty liver diseases are divided into steatosis, steatohepatitis, hepatic cirrhosis, and hepato-carcinoma. Steatosis presents minimal inflammation without hepatocellular injury, but steatohepatitis determines inflammation and hepatocellular injury with cell apoptosis presence, leading to liver fibrosis, cirrhosis, and hepato-carcinoma [11,12,13,14,15,16,17,18]. Chronic hepatic diseases determine dysfunctions in liver metabolism, immune system, liver, and gut barriers. Hepatocytes and immune cells may lose their functions, leading to pro-inflammatory, profibrogenic phenotypes that facilitate cirrhosis progression. Liver cirrhosis is characterized by inflammation and immune deficiency linked to gut dysbiosis [19,20,21,22,23,24,25,26,27].

The gut-derived microbes determine hepatocytes to activate immune surveillance by MHC I/II and costimulatory molecules: LSEC recruit monocytes, lymphocytes, and adhesion molecules [28,29,30,31]. The inflamed liver presents stellate hepatocyte cells differentiated into fibrinogenic and immuno-modulatory cells. Even though the HSCs express the MHC molecules, they indicate poor T lymphocyte activation. Constant exposure to lower levels of LPS determines the endotoxin tolerance installing and programmed death ligand one coinhibitory molecule upregulation. PDL-1 determines IL-10 and TGF-β inhibitory cytokines production and the decreasing of CD4+ and CD8+ T lymphocytes activation [32,33,34,35,36].

Bacterial translocation is bacteria and their metabolite’s movement from the intestinal lumen to the portal bloodstream and mesenteric lymph nodes. BT increases in pathological conditions, determining a pro-inflammatory response [37,38]. In cirrhotic animals, the dysregulated gut immune barrier determines increasing activated lymphocytes and IFN-γ and IL-17 production [37,39]. The inflammation produced by dysbiosis mediated by monocytes and macrophages affects the gut barrier function by the TNF-α/TNFR1 signaling way [1]. In cirrhosis patients, activated macrophages determine NO, IL-6, and IL-8 production under bacterial stimulation, affecting the gut barrier function [38].

The immune system and the hepatic barrier have role in the pathogenesis of ALD and NAFLD. Gut barrier dysfunction by increased intestinal permeability facilitates PAMPs portal influx, determining the hepatic inflammation [40]. Progression of chronic liver diseases is associated with damages in gut defense, leading to a functional dysregulated gut barrier [27].

Our study highlighted the biomarkers pattern to evaluate the immunological dysfunction of the gut barrier. Immunological changes in gut barrier function were evaluated by pro-inflammatory biomarkers such as T helper lymphocytes (CD4), monocytes (CD14), and cytokines (IL-2, IL-6) by flow cytometry methods as the response to immune system activation. Cell death was studied using Annexin V-FITC/propidium iodide dual stain, and oxidative stress by total reactive oxygen species count using flow cytometry methods, to observe endotoxin tolerance installation due to gut immune and liver barrier dysfunctions, and changes in the apoptosis mechanism in gut cells determined by higher oxidative stress leading to pro-inflammatory and profibrogenic phenotypes. Our objective is to study the immune changes by pro-inflammatory biomarkers in gut–liver-axis-linked ROS-cell death mechanisms in chronic and acute inflammation when gut cells are exposed to endotoxins in patients with hepatic cirrhosis or steatosis.

## 2. Results

### 2.1. Pro-Inflammatory Biomarkers in the Gut–Liver Axis

#### 2.1.1. Helper T Lymphocytes and Th1 Cytokines Expressions in Acute or Chronically Intestinal Inflammation Tissue Samples Recovered from Liver Cirrhosis or Hepatic Steatosis Patients

Gut immune barrier alterations in duodenal tissue samples by T lymphocytes and Th1 cytokines, and pro-inflammatory biomarkers using CD4 Alexa Fluor 488 and IL-2 PE stain, were used to establish their different patterns in liver cirrhosis or hepatic steatosis. Patients reported to controls are shown in Figure 1, Figure 2 and Figure 3.

Cell populations double positive (CD4+IL-2+) expressed using Alexa Fluor 488 and PE dual stain had an important role because they offered the possibility to observe changes in the gut immunological barrier function when presented with PAMPs/DAMPs or endotoxins action in cells, which may determine chronically or acute intestinal inflammation (CII or AII) patterns (Figure 1 and Figure 2C).

Inflammation status by CD4+IL-2+ cell population presented significantly increased expressions in CII tissue samples reported to AII tissue samples recovered from patients with hepatic cirrhosis with A-B child (CII: 46.91 ± 4.44 vs. AII: 26.85 ± 2.96 vs. C: 13.13 ± 4.04, *p <* 0.01) or C child stages by Child-Pugh score (CII: 39.70 ± 3.43 vs. AII: 26.82 ± 3.57; *p <* 0.05; vs. C: 13.13 ± 4.04, *p <* 0.01; *p <* 0.05), and controls (C, Figure 1A–F,H–M and Figure 2C).

The dysregulated gut immune barrier was showed by significantly increased values of CD4+IL-2+ pro-inflammatory patterns in CII and AII tissue samples recovered from hepatic steatosis patients reported to controls (CII: 37.15 ± 5.36; AII: 34.57 ± 3.81 vs. C: 13.13 ± 4.04, *p <* 0.05, Figure 1A,D,G,K,M and Figure 2C).

Inflammatory patterns of immune dysregulation in patients with A-B child stage HC were based on statistical increases in activated T lymphocytes (CD4+: CII: 52.23 ± 4.09 vs. AII: 48.37 ± 5.54, *p >* 0.05 vs. C: 30.97 ± 9.15, *p <* 0.05) and IL-2 production (IL-2+: CII: 50.96 ± 3.52 vs. AII: 35.62 ± 3.25; *p <* 0.01; vs. C: 13.93 ± 5.03, *p <* 0.01, Figure 2A,B and Figure 3B,E,I,L), being the results of the innate and adaptative immune system reaction to constant endotoxin exposure of cells in the gut–liver axis.

Instead, in patients with C child stage HC, it was observed that the endotoxin tolerance installed in cells with CD4+ silent activation (CII: 45.53 ± 3.08 vs. AII: 45.32 ± 5.34 vs. C: 30.97 ± 9.15, *p >* 0.05), but there remained statistical increases in Th1 cytokines expression in CII reported to AII or control tissue samples (IL-2+: CII: 42.13 ± 3.93 vs. AII: 25.93 ± 4.13, *p <* 0.05; C: 13.93 ± 5.03, *p <* 0.01, Figure 2A,B and Figure 3C,F,J,M).

Adapted changes in the inflammatory potential of the gut immune environment in hepatic steatosis were observed in CII and AII tissue samples reported to controls (IL-2+: CII: 40.16 ± 6.51; AII: 37.67 ± 4.09 vs. C: 13.93 ± 5.03, *p <* 0.05; CD4+: CII: 46.98 ± 5.74; AII: 59.33 ± 4.34; vs. C: 30.97 ± 9.15, *p >* 0.05, *p <* 0.01, Figure 2A,B and Figure 3D,G,K,N).

#### 2.1.2. Monocytes and Th2 Cytokines Expressions in Acute or Chronically Intestinal Inflammation Tissue Samples Recovered from Liver Cirrhosis or Hepatic Steatosis Patients

Immune cell adaptation from the gut barrier by CD14 monocyte activation, and Th2 cytokines pro-inflammatory releasing, analyzed from different duodenal tissue samples recovered from patients with liver cirrhosis or hepatic steatosis reported to controls, were presented in Figure 4, Figure 5 and Figure 6.

Gut barrier dysfunction and innate immunity reaction represented by CD14+IL-6+ cell populations expressed by Alexa Fluor 488 and phycoerythrin dual stain showed significantly increased values in CII tissue samples than AII tissue samples recovered from patients with A-B child than controls (CII: 47.17 ± 5.52 vs. AII: 31.12 ± 1.70, *p <* 0.05 vs. C: 26.37 ± 2.44, *p >* 0.05; Figure 4A,B,E,I,L and Figure 5C).

In patients with C child stage HC, there was observed a silent increasing immune reaction in experimental samples than controls (CII: 40.53 ± 3.68 vs. AII: 31.88 ± 3.68 vs. C: 26.37 ± 2.44, *p >* 0.05; Figure 4A,C,F,J,M and Figure 5C).

In steatosis patients, CD14+IL-6+ inflammatory patterns of immune dysregulation were represented by significantly increased values in CII and AII tissue samples reported to controls (CII: 37.46 ± 2.36; AII: 41.53 ± 3.32 vs. C: 26.37 ± 2.44, *p <* 0.05, Figure 4A,D,G,K,N and Figure 5C).

In the gut–liver axis, in CII and AII tissue samples recovered from patients with A-B child stage of HC, the immune response mediated by CD14+ monocytes showed higher values (CII: 50.45 ± 4.85 vs. AII: 47.93 ± 3.99, vs. C: 41.64 ± 8.38, *p >* 0.05; Figure 5A and Figure 6A,B,E) and increasing of IL-6 pro-inflammatory releasing (CII: 46.73 ± 4.51; AII: 40.57 ± 2.60 vs. C: 30.83 ± 2.86, *p >* 0.05; *p <* 0.05; Figure 5B and Figure 6H,I,L) than controls.

Inflammatory patterns of immune dysregulation in patients with C child stage of HC showed increasing of activated monocytes (CD14+: CII: 46.83 ± 4.21 vs. AII: 37.86 ± 5.48, vs. C: 41.64 ± 8.38, *p >* 0.05) and IL-6 expressions (IL-6+: CII: 40.96 ± 3.59; AII: 36.57 ± 4.51 vs. C: 30.83 ± 2.86, *p >* 0.05, Figure 5A,B and Figure 6A,C,F,H,J,M) than controls, as a result of the innate immune system reaction to endotoxins action in intestinal cells.

Changes in the gut immune barrier in hepatic steatosis patients were observed in CII and AII tissue samples reported to controls (CD14+: CII: 44.37 ± 2.11; AII: 58.90 ± 4.52; vs. C: 41.64 ± 8.38, *p >* 0.05; IL-6+: CII: 48.73 ± 6.39; AII: 50.59 ± 3.06 vs. C: 30.83 ± 2.86; *p >* 0.05; *p <* 0.01; Figure 5A,B and Figure 6A,D,G,H,K,N).

### 2.2. Cell Apoptosis in the Gut–Liver Axis

Apoptosis and necrosis, two major types of cell death, were studied using Annexin V-FITC/PI dual stain in different duodenal tissue samples recovered from patients with liver cirrhosis and hepatic steatosis reported to controls, as presented in Figure 7 and Figure 8.

The cell viability showed significant differences between chronic intestinal inflammation and acute intestinal inflammation samples in patients with hepatic cirrhosis (HC) with A-B child stages using Child-Pugh score than controls (CII: 41.92 ± 6.96 vs. AII: 87.49 ± 2.44, *p <* 0.01 vs. C: 98.21 ± 0.29, *p <* 0.01; *p <* 0.05, Figure 7A,B,E and Figure 8A).

Patients with hepatic cirrhosis (HC) with C child stage relived significantly lower cell viability values in CII than AII tissue samples reported to controls (CII: 37.79 ± 4.12 vs. AII: 81.59 ± 2.79, *p <* 0.01 vs. C: 98.21 ± 0.29, *p <* 0.01, Figure 7A,C,F and Figure 8A).

Changes in the gut immune barrier in hepatic steatosis patients determine a significant decrease of cell viability in CII than AII tissue samples reported to controls (CII: 44.29 ± 2.19 vs. AII: 78.99 ± 4.24, *p <* 0.01 vs. C: 98.21 ± 0.29, *p <* 0.01, *p <* 0.05, Figure 7A,D,G and Figure 8A).

Cell apoptosis, characterized by biochemical and morphological changes, showed significantly increasing values in CII than AII tissue samples in patients with hepatic cirrhosis with A-B child (CII: LA: 41.32 ± 6.71 vs. AII: 0.27 ± 0.77, *p <* 0.01, CII: EA: 8.22 ± 3.87 vs. AII: 0.26 ± 0.73, *p* < 0.01) or C child stages (CII: LA: 42.16 ± 17.63 vs. AII: 0.41 ± 0.69, *p* < 0.01, CII: EA: 9.90 ± 8.42 vs. AII: 0.25 ± 0.52, *p* < 0.01, Figure 7A–C,E,F and Figure 8B,C).

In hepatic steatosis patients, there were observed significantly higher values of cell apoptosis in CII than AII tissue samples (CII: LA: 33.92 ± 4.86 vs. AII: 0.00 ± 0.00, *p <* 0.01, CII: EA: 12.02 ± 7.19 vs. AII: 0.00 ± 0.00, *p <* 0.01, Figure 7A,D,G and Figure 8B,C).

Necrosis of gut cells presented significant differences between CII and AII tissue samples in patients with hepatic cirrhosis with A-B child (CII: 8.51 ± 1.71; AII: 11.74 ± 2.21 vs. C: 1.78 ± 0.29, *p <* 0.05) or C child stages reported to controls (CII: 10.16 ± 2.40; AII: 17.73 ± 2.60 vs. C: 1.78 ± 0.29, *p >* 0.05; *p <* 0.01; Figure 7A–C,E,F and Figure 8D).

In hepatic steatosis patients, necrosis showed significantly lower values in CII than in AII tissue samples and controls (CII: 9.88 ± 2.84; AII: 20.99 ± 4.24 vs. C: 1.78 ± 0.29, *p >* 0.05; *p <* 0.01; *p <* 0.05, Figure 7A,D,G and Figure 8D).

### 2.3. Oxidative Stress in the Gut–Liver Axis

In different duodenal cell samples recovered from patients with hepatic chronic diseases reported to controls. The oxidative cellular stress analyzed by total ROS is presented in Figure 9.

In patients with hepatic cirrhosis (HC) with A-B child, oxidative stress presented significantly increasing values in CII than AII tissue samples and controls (CII: 40,591.66 ± 21,719.46 × 10^4^; AII: 4697.50 ± 155.15 × 10^4^ vs. C: 371.51 ± 343.14 × 10^4^, *p <* 0.01) or C child stages (CII: 17,572.22 ± 9315.62 × 10^4^; AII: 3456.70 ± 1225.84 × 10^4^ vs. C: 371.51 ± 343.14 × 10^4^, *p <* 0.01; *p >* 0.05; Figure 9A–D,G).

ROS levels in hepatic steatosis patients showed significantly higher values in CII than in AII tissue samples and controls (CII: 66,266.66 ± 21,422.08 × 10^4^ vs. AII: 12,751.25 ± 8594.51 × 10^4^, *p <* 0.05; vs. C: 371.51 ± 343.14 × 10^4^, *p <* 0.05; Figure 9A,E–G).

Figure 10A–L summarizes our main findings about immune changes in the gut–liver-axis-linked ROS-cell death mechanisms in chronic and acute inflammation when gut cells are exposed to endotoxins.

Late apoptosis represents the chronic response to injury induction by gut immune barrier dysfunction, oxidative stress, and the liver-dysregulated barrier, being observed in duodenal tissue samples with chronic inflammation recovered from patients with hepatic cirrhosis and steatosis, compared with controls, using Annexin V-FITC/PI dual stain (Figure 10H,J).

In the meantime, necrosis represents an acute and severe reply to endotoxin action in gut cells when the innate and adaptive functional immune system reacts to pro-inflammatory Th1 and Th2 cytokines releasing, offering protection against PAMPs/DAMPs by monocytes and T lymphocytes activation, being observed in duodenal tissue samples with acute inflammation recovered from patients with hepatic cirrhosis and steatosis, compared with controls, using Annexin V-FITC/PI dual stain (Figure 10G,I).

An interesting oxidative stress mechanism was observed in CII reported to AII tissue samples of hepatic chronic diseases, in gut cells, there is a constant action of endotoxins, which determines the installing of the endotoxin tolerance because of gut immune and liver barrier dysfunctions. Higher oxidative stress in gut cells determines changes in the apoptosis mechanism, leading to pro-inflammatory and profibrogenic phenotype status (Figure 10K,L).

## 3. Discussion

The gut barrier is a morphological functional mechanism that includes epithelial, immunological, vascular, and liver barriers [41].

Our study presents the adapted changes in the inflammatory potential of the gut immune environment highlighted by T lymphocytes, monocyte activation, and IL-2 production in patients with chronic hepatic diseases. The significant innate and adaptative immune system reaction resulting from constant endotoxin exposure of gut cells in chronic inflammation tissue samples recovered from hepatic cirrhosis with A-B child stage highlights the dysregulated gut immune barrier function. Instead, in patients with C child stage of HC, the endotoxin tolerance was installed in cells, characterized by T lymphocyte silent activation and increased Th1 cytokines expression.

Non-alcoholic fatty liver disease presents an increased risk of causing advanced liver diseases. Mechanisms of the intestinal barrier and permeability are disrupted in NAFLD. The gut barrier function is based on the microbiome integrity, mucus, enterocytes, immune cells, and vascular barrier [42,43,44,45]. The immune barrier includes Paneth cells, and B and T lymphocytes [46]. The epithelial barrier is the innate immunity site being formed by enterocytes producing defensins, goblet cells secreting mucus, tuft cells releasing IL-25 and IL-13, Paneth cells producing defensins, and M cells inducing secretory immunoglobulin A [47,48,49,50,51]. Innate immune cells are involved in metabolic homeostasis, releasing cytokines, and preceding the adaptive T lymphocytes response [52,53]. Receptors with a role in PAMPs and DAMPs recognizing are implied in dendritic cell recruitment when the gut barrier is altered. Dendritic cells transport antigens useful to B and T lymphocyte maturation [53,54]. The Kupffer cells have essential roles in liver barrier maintenance, phagocytizing the bacteria from the bloodstream, and eliminating the PAMPs and endotoxins. The Kupffer cells activation depends on human lipopolysaccharide–lipopolysaccharide-binding protein-complex-stimulating myeloid cells by CD14 monocytes and Toll-like receptor 4 [55,56,57].

Low-grade chronic inflammation determines the metabolic progression from NAFL to NASH to cirrhosis [58,59,60]. Immune cell adaptation affects gut permeability, determining bacterial translocation. Modified gut microbiome contributes to inflammatory and fibrosis responses in NAFLD patients [61]. Another study confirmed that gut permeability was affected in NASH patients. The bacteria overgrowth in small intestines determines increasing CD14+ monocytes and IL-8 expressions [62]. In hepatic cirrhosis, TLR4-LPS interaction determines TNF, IL-1, IL-6 pro-inflammatory cytokines, and chemokines production [32,63].

Hepatic macrophages are divided into Kupffer cells and monocyte-derived macrophages with roles in maintaining immune homeostasis. Macrophages contribute to cirrhosis progression by promoting inflammation and fibrogenesis. DAMPs and PAMPs activate the macrophages, secreting the TNF, IL-1β, IL-6, IL-8, and ROS pro-inflammatory mediators, determining the HSCs activation using TGF-β1 and PDGF signaling [64]. Macrophage-derived inflammasome resulting from bacterial translocation contributes to hepatic inflammatory injury. Activated inflammasome initiates caspase-1 and determines IL-1β and IL-18 pro-inflammatory cytokines production, increasing liver inflammation, fibrosis, and damage in ADL and NAFLD [65,66,67].

In liver cirrhosis, the presence of PAMPs and DAMPs associated with the altered gut barrier determines the transformation of tolerogenic properties of the liver into immunogenic and fibrinogenic properties based on the expansion of pro-inflammatory cells and cytokines from the extracellular matrix [68]. The PAMPs binding to PRRs in tissues determines the immune cells’ activation and pro-inflammatory cytokines’ release. The presence of bacteria determines cell death when oxidative stress overwhelms the processing abilities of the endoplasmic reticulum, leading to unfolded protein responses determining the production of IL-6 and TNF pro-inflammatory cytokines. Positive feedback is assured by the IL-1, IL-6, IL-8, and TNF systemic cytokines using UFR activation in the liver, increasing systemic and hepatic inflammation [69,70,71]. A persistent inflammation determines parenchymal and systemic immune cell apoptosis. In cirrhosis, CD14+CD16+ monocytes increase, expressing more TNF, determining a pro-inflammatory and profibrogenic phenotype [72,73]. A recent study showed that in response to bacterial invasion, IL-2 pro-inflammatory cytokine secreted by immune cells affects T helper cells and decreases humoral immunity in advanced cirrhosis [74].

PAMPs are produced by the interaction of the microbiota with endogenous and exogenous substances, including gases, metabolites, and bacterial products. PAMPs and TLRs interaction activates intracellular molecular pathways determining the NF-kB, TNF-α, IL-1β, IL-6, IL-12, and IL-18 cytokines activation and nitric oxide production [75]. T lymphocytes, neutrophils, and monocytes determine pro-inflammatory changes [32,76], apoptosis, and necrosis [77]. The production of reactive oxygen species contributes to liver damage [75,78]. Kupffer cells activation by ROS presence determines ROS, cytokines, and chemokines production [63,79,80]. Gut mucosa mechanisms act IL-10 releasing, determining a decrease in endotoxin absorption [80,81]. In gut barrier with increased permeability, enterotoxins from portal circulation determine pro-inflammatory modifies by action on hepatic pattern-recognition receptors [75,82].

Hepatic stellate cell activation is responsible for NASH progression by TLR4 signaling way by increasing TNF-α expression [75,83], determining a release of pro-inflammatory cytokines and oxidative stress [84]. In steatosis mice models, the presence of endotoxins triggers liver inflammation. Instead, in obese mice models, there were observed in portal circulation high levels of endotoxins and IL-1, IL-6, INF-γ, and TNF-α pro-inflammatory cytokines reported to control mice. HSCs were activated by enhanced sensitivity to LPS and increased cytokines levels, determining a dysregulated gut barrier [85].

NAFLD patients showed small intestinal bacterial overgrowth [86,87], dysbiosis [88], and abnormal gut permeability, leading to liver inflammation and fibrosis [89,90,91].

In advanced cirrhosis, IL-2 pro-inflammatory cytokine is secreted by cells as a response to bacteria translocation [20,74]. In patients with alcoholic hepatitis, there were observed increasing levels of TNF-α, IL-6, IL-8, and IL-18 pro-inflammatory cytokines. IL-6 protects against apoptosis [92], reducing alcoholic liver injury and inflammation [93], and has a protective effect in the ALD early phase [94].

Our study observed interesting ROS-cell death mechanisms in chronic and acute inflammation samples when gut cells are exposed to endotoxins and immune changes in the gut–liver axis. Late apoptosis represents the chronic response to injury induction by the gut immune barrier dysfunction, oxidative stress, and dysregulated liver barrier. Meanwhile, necrosis represents an acute and severe reply to endotoxin action on gut cells when the innate and adaptive functional immune system reacts to pro-inflammatory Th1 and Th2 cytokines releasing, offering protection against PAMPs/DAMPs by monocytes and T lymphocyte activation. An interesting oxidative stress mechanism was observed in chronic and acute inflammation samples of hepatic chronic diseases when there exists a constant action of endotoxins in gut cells, which determines the installation of endotoxin tolerance due to gut immune and liver barrier dysfunctions. Higher oxidative stress in gut cells determines changes in the apoptosis mechanism, leading to pro-inflammatory and profibrogenic phenotype status.

A significant source of reactive oxygen species in normal conditions is represented by mitochondria. TNF-α released from Kupffer cells stimulated by endotoxins leads to a decrease in mitochondrial complex III function [95]. ROS is implied in the pro-inflammatory process. In the injured liver, pro-inflammatory cytokines and ROS produced by macrophages and infiltrating leukocytes determine HSCs transformation in activated phenotype, responsible for fibrosis, cirrhosis, and cancer development [96,97]. Hepatic fibrosis is preceded by chronic inflammation. Liver inflammation is associated with necrosis and apoptosis of hepatocytes. In patients with liver cirrhosis, there were observed high levels of systemic IL-6 with a dysregulated acute phase response of the immune system [98]. Another study reported that IL-1α and IL-6 serum levels were significantly increased in alcoholic cirrhosis patients, being correlated with the progression of liver cirrhosis [99].

Oxidative stress represents an essential aspect of research linked to intra- and extrahepatic disorders produced by NAFLD [100,101,102]. The oxidative stress mechanisms are linked to mitochondria and endoplasmic reticulum dysfunctions determining hepatic structure and function damages. These alterations of liver tissue by ROS affect extrahepatic tissues [103]. In the liver, triglycerides and FFA induce lipotoxicity and oxidative stress, determining inflammation, mitochondrial dysfunction, apoptosis, and fibrosis. The progressive hepatocyte death by high oxidative stress promotes cirrhosis and HCC [104,105]. In NAFLD, β-oxidation dysfunction determines the increasing of liver lipids levels and inflammation [106], leading to the pro-inflammatory and apoptotic response of the immune system [107,108]. Activation of Kupffer cells by ROS triggers pro-inflammatory cytokines and chemokines released by macrophages [109,110].

In liver diseases, there are two significant types of cell death. Apoptosis is an early, chronic response to injury induction, whereas necrosis is an acute and severe reply. Biochemical mechanisms and morphological changes characterize apoptotic cells, including death receptor and mitochondria-dependent pathways [111,112,113].

Long-term use of alcohol leads to alcoholic liver disease development [114]. Alcohol stimulates ROS production and leads to apoptosis via the oxidative stress mechanism. Alcohol’s impact on hepatocytes includes endoplasmic reticulum stress and mitochondrial dysfunction [115]. Alcohol alters endotoxin receptors, determining KCs’ tolerance to endotoxins. Alcohol activates KCs to be sensitized by LPS via TLR4, promoting TNFα and ROS production [116]. Pro-inflammatory factors determine liver dysfunction, apoptosis, necrosis, and fibrosis. The pro-inflammatory response of immune cells by increased ROS levels contributes to HSC activation [117]. The activated HSCs’ response to recurrent hepatic injury determines extra-cellular matrix protein accumulation, especially collagen type I. Apoptosis, inflammation, and fibrosis are characteristics of ADL [118].

NAFLD’s pathological changes are related to dysregulated lipid metabolism and chronically pro-inflammatory-oxidative stress response [119]. Macrophages and adipocytes secrete IL-6 and TNFα pro-inflammatory cytokines [120]. ROS are neutralized by mitochondrial uncoupling protein two upregulation and limited synthesis of mitochondrial adenosine triphosphate [121]. A lower antioxidant response induces sensitization of cells to mitochondrial and cellular apoptotic damages. FFAs activate the apoptosis pathway by Bim and Bax, determining mitochondrial permeabilization, cytochrome c release, and caspases activation [122,123]. Pro-inflammatory cytokines from gut sources sensitize the liver to ROS and cellular lesions [124,125].

Interleukins assure the pro-inflammatory response of immune cells in their interaction with target cells [126]. A higher systemic IL-6 level is a diagnostic biomarker to detect inflammatory conditions [127]. Patients with acute and chronic liver diseases present increased IL-6 pro-inflammatory cytokines levels [128]. In experimental obese mice models, IL-6 way signaling induced by TNFα lower level determined liver inflammation and carcinogenesis [129]. In patients with chronic liver diseases, IL-6 levels in serum and liver tissue are biomarkers of disease progression [128,130]. In liver cancer, IL-6 presents a protective role in fibrogenesis. Kupffer cells stimulate the releasing of pro-inflammatory cytokines, which are implied in tissue remodeling and fibrosis [131]. Cytokines are implied in the regulation of inflammatory response and homeostasis [132]. IL-6 has roles in cell differentiation and apoptosis blocking by modulating the specific gene transcription in hepatic inflammatory processes. IL-6 and transforming growth factor β induce IL-17 release from T lymphocytes. IL-2, IL-15, IL-18, and IL-21 pro-inflammatory cytokines stimulate IL-17 production from activated T lymphocytes [133].

Macrophages are implied in chronic liver lesion pathogenesis [134]. In inflammation and fibrosis, macrophage receptors activate downstream molecules by different signaling pathways [135,136,137]. In healthy conditions, macrophages do not promote fibrotic responses when liver cell apoptosis happens daily. There are observed inflammatory responses and liver fibrosis when hepatocyte necrosis appears [138,139,140,141,142,143]. Hepatic macrophages promote fibrosis through immune cell recruitment and pro-inflammatory cytokines and chemokines secretion in the early stages [144]. M1 and M2 macrophages, helper T lymphocyte responses express CD68, CD163, and CD14 [145,146]. M1 cells determine Th1 response by TNF-α, IL-1, and IL-12 pro-inflammatory cytokines secretion, and reactive oxygen species, promoting inflammation, and liver fibrogenesis [147,148]. M2 cells determine Th2 response by TGF-β1, IL-4, IL-13, and IL-10 immuno-modulatory cytokines secretion induced by IL-4 and IL-13 production [149]. Patients with hepatitis and cirrhosis presented increased CD14 +CD16+ macrophages in liver tissue samples. M1 macrophages promote IL-1, IL-6, and IL-23 pro-inflammatory cytokines secretion, the differentiation of Th17 cells, the determination of lymphocytes infiltration, and cell death [146,147,149].

The interactions between the liver and gut, mediated by the immune system, determine the liver transition from immune-tolerant to immune-active status, with TGF-β, IL-1, IL-6, and TNF-α pro-inflammatory cytokines production. A high oxidative stress resulting from the presence of endotoxin represents another mechanism for inducing liver damage and inflammation [150].

Going over the limitations of the study, such as a small number of patients or small pieces of duodenal biopsy, flow cytometric analysis of pro-inflammatory biomarkers, and apoptosis linked to oxidative stress mechanisms, may be used as a direction for future research in different maligned affections as efficient means of measurement in the diagnostic field. [151]. In this paper, we studied only the immunological changes in the gut barrier by pro-inflammatory biomarkers linked to ROS-cell death mechanisms, not the entire gut barrier function in the gut–liver axis. The bacteria and endotoxins translocation from the liver to the gut and from the gut to the liver due to dysregulated hepatic barrier (patients with cirrhosis or steatosis) determine a pro-inflammatory and profibrogenic phenotype not only in the liver (already cirrhosis or steatosis presence) but also in gut cells being highlighted by ROS-cell apoptosis mechanisms. In function of immune response of patients determine an acute inflammation (when existing a functional immune response) or chronic inflammation (when gut cells present a modified phenotype in a pro-inflammatory and profibrogenic environment characterized by late apoptosis presence and high ROS levels).

## 4. Materials and Methods

### 4.1. Samples Selection

Duodenal biopsies (n = 116) were recovered from patients from the Gastroenterology Department, “Sf. Apostol Andrei” Clinical Emergency County Hospital, Constanta, Romania. Experimental groups were formed by patients with liver cirrhosis (n = 58) with acute or chronic intestinal inflammation (AII; CII) and patients with hepatic steatosis (n = 42) with acute or chronic intestinal inflammation (AII; CII). Experimental intestinal tissue samples were reported to control samples recovered from healthy patients.

Tissue samples were homogenized with TissueRuptor II equipment (Qiagen, Hilden, Germany), being analyzed by flow cytometry methods (CD4 lymphocytes, CD14 monocytes, IL-2, IL-6 cytokines, cell apoptosis, and total reactive oxygen species) in The Cell Biology Department, CEDMOG, Ovidius University from Constanta, Romania.

### 4.2. Equipment and Reagents

Our study used Attune, an Acoustic focusing cytometer (Applied Biosystems, part of Life Technologies, Bedford, MA, USA), setting with Attune performance tracking beads, labeling, and detection (Life Technologies, Europe BV, Bleiswijk, The Netherlands) [152]. Forward Scatter and Side Scatter gated 10,000 cells per sample for each analysis. Flow cytometry results were collected by Attune Cytometric Software v.1.2.5, Applied Biosystems, 2010. The FITC Annexin V/Dead Cell Apoptosis Kit (Life Technologies, Europe BV, Bleiswijk, The Netherlands) was used to observe the apoptotic cells. Anti-Hu CD4 Alexa Fluor 488 (Clone OKT4, Life Technologies, Europe BV, Bleiswijk, The Netherlands), anti-Hu CD14 Alexa Fluor 488 (Clone 61D3, Life Technologies, Europe BV, Bleiswijk, The Netherlands), anti-Hu IL-2 PE (Clone MQ1-17H12, Life Technologies, Europe BV, Bleiswijk, The Netherlands), and anti-Hu IL-6 PE (Clone MQ2-13A5, Life Technologies, Europe BV, Bleiswijk, The Netherlands) monoclonal antibodies were used to assess T helper lymphocytes (Th), monocytes, and cytokines secreted by Th1 cells (IL-2) and Th2 cells (IL-6). The negative control was represented by mouse IgG1 kappa Isotype PE (Clone P3.6.2.8.1, Life Technologies, Europe BV, Bleiswijk, The Netherlands). The Total Reactive Oxygen Species Assay Kit 520 nm (Life Technologies, Europe BV, Bleiswijk, The Netherlands) was used to study the cellular oxidative stress.

### 4.3. Surface Glycoproteins of Leucocytes and Cytokines Analysis

An amount of 100 µL duodenal homogenized cells were used for each tube: 1. CD4 Alexa Fluor 488 and IL-2 PE dual stain; 2. CD14 Alexa Fluor 488 and IL-6 PE dual stain; 3. control negative-IgG1 stain.

In total, 5 µL of CD4 Alexa Fluor 488 and 5 µL of IL-2 PE were introduced in tubes with gut cells. An amount of 5 µL of CD14 Alexa Fluor 488 and 5 µL of IL-6 PE were added in other experimental tubes with cells. Control tubes with cells were realized for each experimental tube, and 5 µL of the mouse IgG1 negative control was added. All samples were vortexed and incubated for 25 min. at room temperature in darkness. In total, 1 mL of Flow Cytometry Buffer (Life Technologies, Europe BV, Bleiswijk, The Netherlands) was added in tubes. Glycoproteins and cytokines were identified by flow cytometry methods using the size and specificity of CD4, CD14, IL-2, and IL-6 expressions, and were analyzed with BL1 channel for Alexa Fluor 488 and BL2 channel for PE.

### 4.4. Cell Death Analysis

In total, 200 µL duodenal homogenized cells were introduced in flow cytometry tubes. An amount of 2 µL of Annexin V-FITC and 2 µL PI were added in tubes and kept in darkness for 30 min at room temperature. An amount of 1 mL of Flow Cytometry Buffer was added after incubation. Samples were analyzed by BL1 channel to 488 nm excitation, green emission for Annexin V-FITC, and BL2 channel in orange emission for propidium iodide.

### 4.5. Total ROS Activity Assay

In flow cytometry tubes, 50 µL of ROS Assay Stain 1x solution and 500 µL homogenized cell samples were mixed well. Samples were incubated at 37 °C in 5% CO_2_ conditions for 60 min. Samples were analyzed by flow cytometry methods using the BL1 channel in green emission and 488 nm excitation for ROS.

### 4.6. Data Analysis

Our results are presented as means values ± standard error (SE), representing our flow cytometry methods as pro-inflammatory biomarkers expressions such as CD4+ T helper lymphocytes (%), Th1 and Th2 cytokines releasing (IL-2, IL-6, %), CD14 monocytes (%), apoptosis and necrosis (%), and count (×10^4^) of oxidative stress (Total ROS). For a normal distribution, the Kolmogorov–Smirnov test was used, while differences between CII and AII and control tissue samples were calculated with an independent t-test, *p <* 0.05 being considered statistically significant, by MedCalc v20.111 Software Ltd. (Ostend, Belgium). Figure 1, Figure 3, Figure 4, Figure 6, Figure 7, and Figure 9A–F and Figure 10 were made with Attune Cytometric Software v.1.2.5, Applied Biosystems, 2010 (Bedford, MA, USA), and Figure 2, Figure 5, Figure 8, and Figure 9G were made with MedCalc v20.111 Software Ltd. (Ostend, Belgium).

## 5. Conclusions

Our study highlighted ROS-cell death mechanisms in chronic or acute inflammation when exposure of gut cells to endotoxins and immune changes in the gut–liver axis.

Late apoptosis is a chronic response to injury induction by gut immune barrier dysfunction, oxidative stress, and the liver-dysregulated barrier. In contrast, necrosis is an acute and severe reply to endotoxin action on gut cells when the innate and adaptive functional immune system reacts to pro-inflammatory Th1 and Th2 cytokines releasing, offering protection against PAMPs/DAMPs by monocytes and T lymphocytes activation in patients with hepatic cirrhosis and steatosis.

A persistent endotoxin action on gut cells determines an increase in oxidative stress, and endotoxin tolerance is installed due to gut immune and liver barrier dysfunctions. A higher oxidative stress in gut cells determines changes in the apoptosis mechanism, leading to pro-inflammatory and profibrogenic phenotype status.

## Figures and Tables

**Figure 1 ijms-25-02472-f001:**
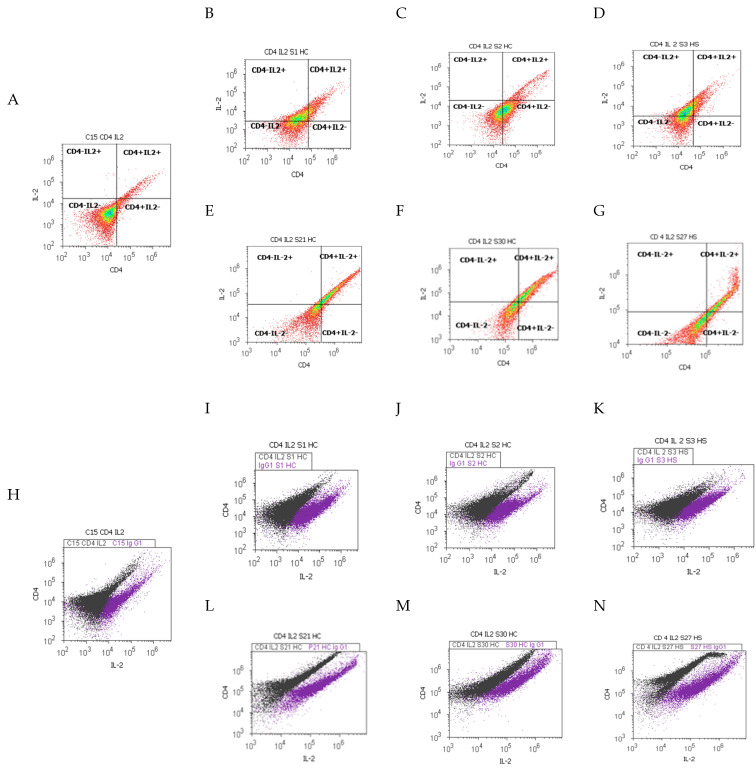
Surface T cell glycoproteins and Th1 cytokines expressions highlighted by the double-positive cell populations (CD4+IL-2+) with CD4 Alexa Fluor 488 and IL-2 PE dual stain. ***Inflammation status by CD4+IL-2+ cell population:*** (**A**) 5.24%; (**B**) 16.61%; (**C**) 8.81%; (**D**) 33.67%; (**E**) 40.69%; (**F**) 37.80%; (**G**) 27.25%. **Legend:** (**A**,**H**) control duodenal tissue sample C recovered from the healthy patient; (**B**–**N**) experimental duodenal tissue samples (S) recovered from patients with hepatic cirrhosis (HC) with A child (**B**,**E**,**I**,**L**) or C child (**C**,**F**,**J**,**M**) stages by Child-Pugh score or hepatic steatosis (HS; **D**,**G**,**K**,**N**) with acute intestinal inflammation (AII; **B**–**D**,**I**–**K**) or chronic intestinal inflammation (CII; **E**–**G**,**L**–**N**); (**H**–**N**)-negative control (IgG1) extrapolated on graphic.

**Figure 2 ijms-25-02472-f002:**
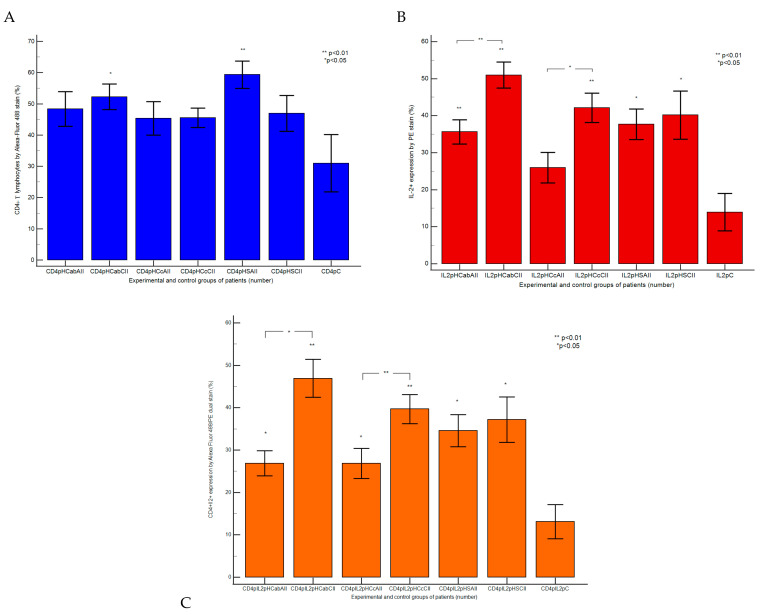
Pro-inflammatory glycoproteins pattern statistics: (**A**) CD4+ Helper T cells population expressed by Alexa Fluor 488 stain; (**B**) Interleukine-2 (IL-2+, Th1 cytokine) expression by PE stain; (**C**) double positive expressions of CD4 T helper cells and IL-2 by Alexa Fluor 488/PE dual stain. **Legend:** HC—hepatic cirrhosis; HS—hepatic steatosis; C—control; ab; c—Child-Pugh score; CII; AII—chronic or acute intestinal inflammation; CD4p—T helper lymphocytes positive populations; IL-2p—interleukine-2 positive expression.

**Figure 3 ijms-25-02472-f003:**
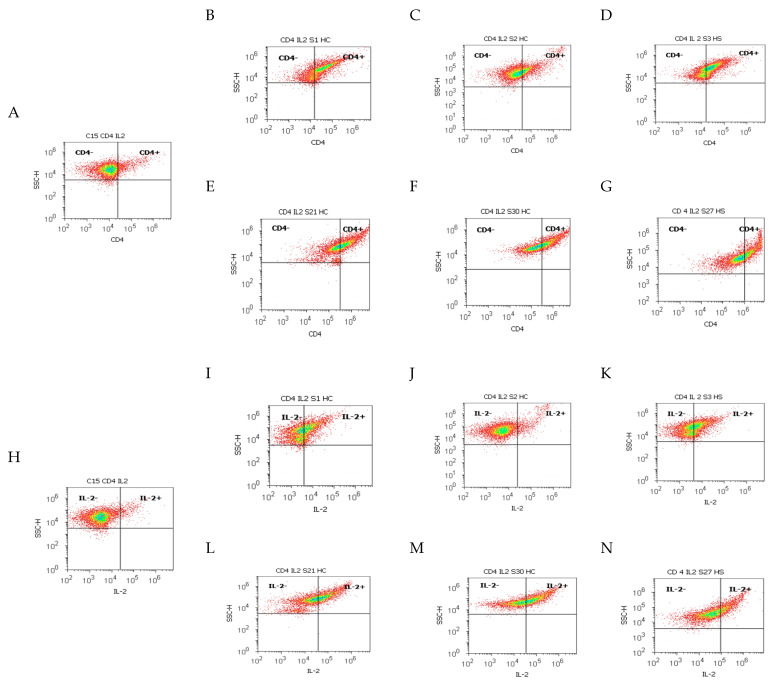
Pro-inflammatory biomarkers. CD4 Helper T cells positive or negative populations expressed by Alexa Fluor 488 stain, Interleukine-2 (IL–2) positive or negative expressions by PE stain. **Legend:** (**A**,**H**) healthy patient C; (**B**–**N**) patients with hepatic cirrhosis (HC) with A child (**B**,**E**,**I**,**L**) or C child (**C**,**F**,**J**,**M**,**J**) stages by Child-Pugh score or hepatic steatosis (HS; **D**,**G**,**K**,**N**) with acute intestinal inflammation (**B**–**D**,**I**–**K**) or chronic intestinal inflammation (**E**–**G**,**L**–**N**).

**Figure 4 ijms-25-02472-f004:**
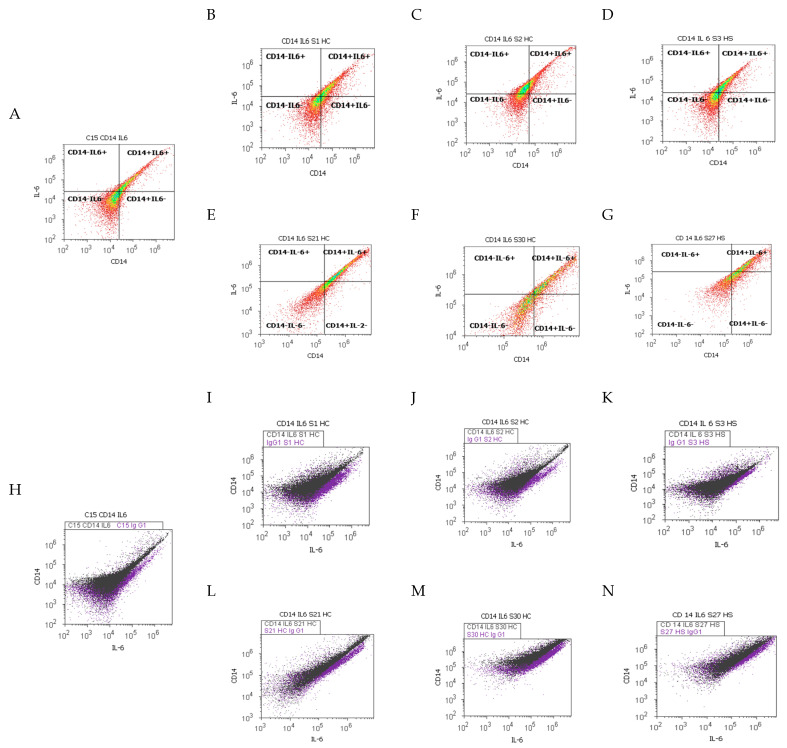
Monocytes and Th2 cytokines expressions highlighted by the double-positive cell populations (CD14+IL-6+) with CD14 Alexa Fluor 488 and IL-6 PE dual stain. ***Inflammatory reaction by CD14+IL-6+ cell population:*** (**A**) 22.47%; (**B**) 28.41%; (**C**) 21.13%; (**D**) 30.65%; (**E**) 42.44%; (**F**) 31.55%; (**G**) 32.87%. **Legend:** (**A**,**H**) control duodenal tissue sample C recovered from the healthy patient; (**B**–**N**) experimental duodenal tissue samples (S) recovered from patients with hepatic cirrhosis (HC) with A child (**B**,**E**,**I**,**L**) or C child (**C**,**F**,**J**,**M**) stages by Child-Pugh score or hepatic steatosis (HS; **D**,**G**,**K**,**N**) with acute intestinal inflammation (AII; **B**–**D**,**I**–**K**) or chronic intestinal inflammation (CII; **E**–**G**,**L**–**N**); (**H**–**N**) negative control (IgG1) extrapolated on graphic.

**Figure 5 ijms-25-02472-f005:**
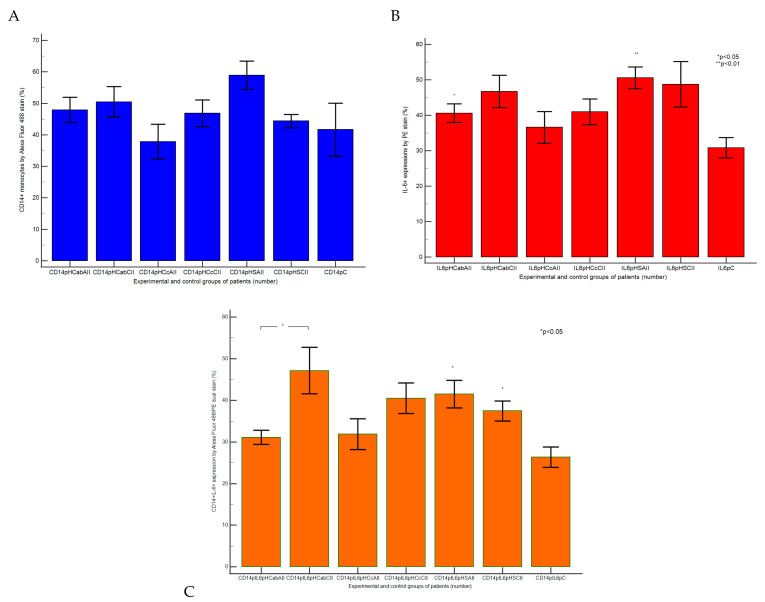
Immune reaction statistics: (**A**) CD14 monocytes positive populations expressed by Alexa Fluor 488 stain; (**B**) Interleukine-6 (IL-6, Th2 cytokine) positive expression by PE stain; (**C**) double positive expressions of CD14 monocytes and IL-6 by Alexa Fluor 488/PE dual stain. **Legend:** HC—hepatic cirrhosis; HS—hepatic steatosis; C—control; ab; c—Child-Pugh score; CII; AII—chronic or acute intestinal inflammation; CD4p—T helper lymphocytes positive populations; IL-2p—positive expression by PE stain.

**Figure 6 ijms-25-02472-f006:**
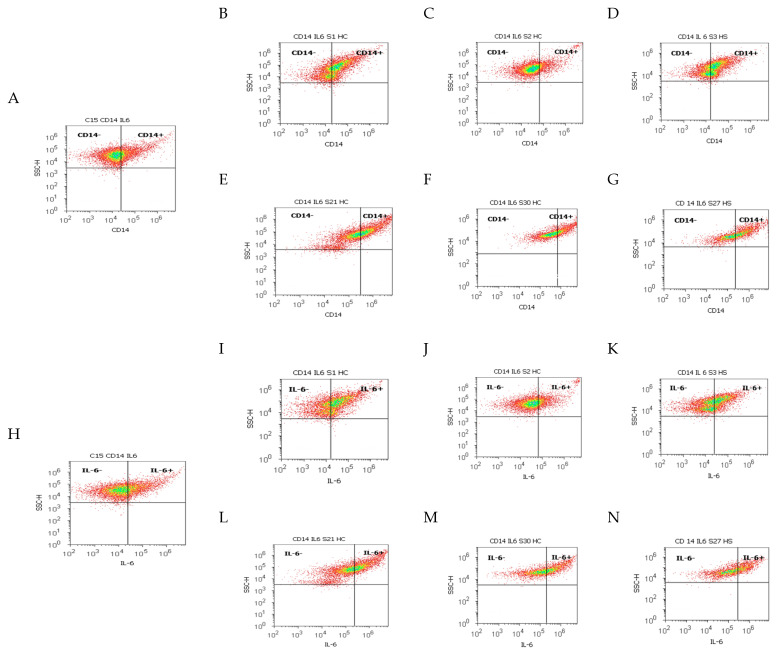
Pro-inflammatory biomarkers. T lymphocytes (monocytes) positive or negative cell populations expressed using Alexa Fluor 488 stain; Interleukine-6 (IL-6) positive or negative expressions by PE stain. **Legend:** (**A**,**H**) healthy patient C; (**B**–**N**) patients with hepatic cirrhosis (HC) with A child (**B**,**E**,**I**,**L**) or C child (**C**,**F**,**J**,**M**,**J**) stages by Child-Pugh score or hepatic steatosis (HS; **D**,**G**,**K**,**N**) with acute intestinal inflammation (**B**–**D**,**I**–**K**) or chronic intestinal inflammation (**E**–**G**,**L**–**N**).

**Figure 7 ijms-25-02472-f007:**
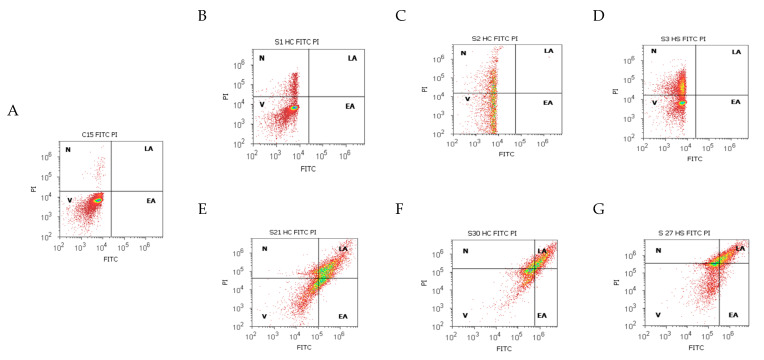
Cell death by Annexin V-FITC/propidium iodide (PI) dual stain. ***Viability (V):*** (**A**) 99.08%; (**B**) 91.21%; (**C**) 88.87%; (**D**) 63.59%; (**E**) 33.33%; (**F**) 41.67%; (**G**) 43.30%. ***Late apoptosis (LA):*** (**A**) 0.00%; (**B**) 0.00%; (**C**) 0.005%; (**D**) 0.00%; (**E**) 46.23%; (**F**) 39.61%; (**G**) 33.42%. ***Necrosis (N):*** (**A**) 0.91%; (**B**) 8.79%; (**C**) 11.11%; (**D**) 36.41%; (**E**) 8.09%; (**F**) 14.83%; (**G**) 17.27%. (**A**) healthy patient C; (**B**,**C**,**E**,**F**) patients with hepatic cirrhosis (HC) with A child (**B**,**E**) or C child (**C**,**F**) stages by Child-Pugh score or hepatic steatosis (HS; **D**,**G**) with acute intestinal inflammation (AII; **B**–**D**) or chronic intestinal inflammation (CII; **E**–**G**).

**Figure 8 ijms-25-02472-f008:**
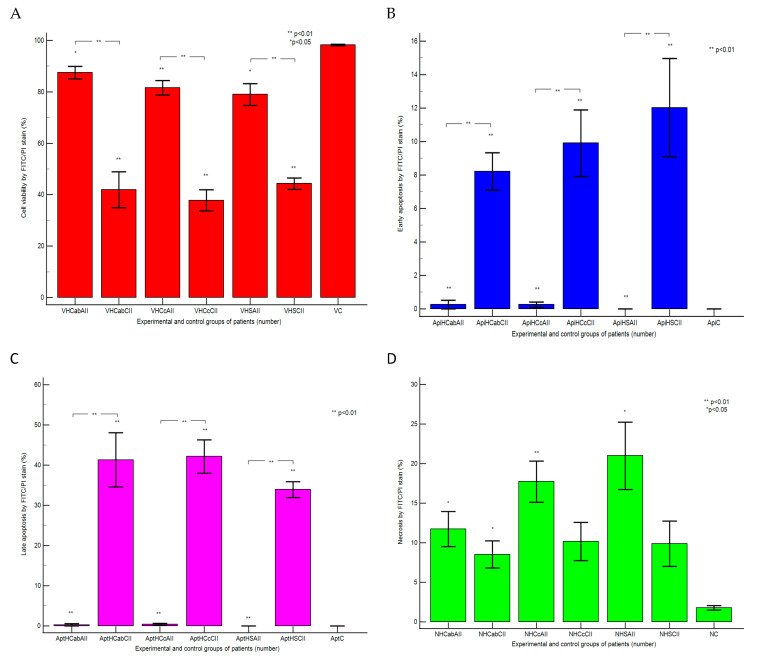
Apoptosis statistics: Cell viability (**A**), early apoptosis (**B**), late apoptosis (**C**), and necrosis (**D**) using Annexin V-FITC/propidium iodide (PI) dual stain. **Legend:** HC—hepatic cirrhosis; HS—hepatic steatosis; C—control; ab; c—Child-Pugh score; CII; AII—chronic or acute intestinal inflammation; V—viability; Api—early apoptosis; Apt—late apoptosis; N—necrosis.

**Figure 9 ijms-25-02472-f009:**
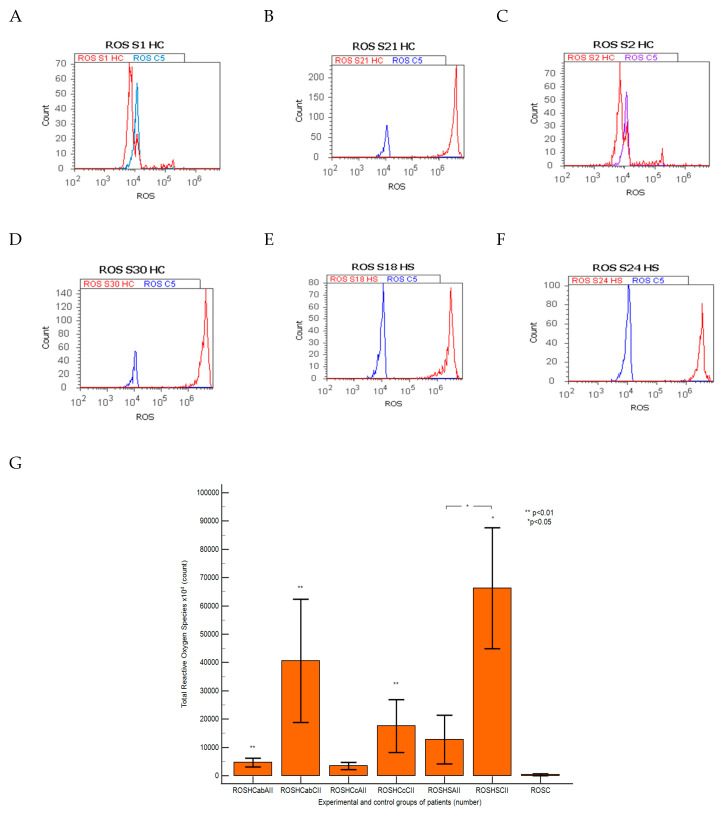
Total reactive oxygen species pattern (ROS). ***ROS:*** (**A**) 70 × 10^5^; (**B**) 200 × 10^7^; (**C**) 50 × 10^6^; (**D**) 142 × 10^7^; (**E**) 72 × 10^7^; (**F**) 80 × 10^7^ reported to control tissue sample (C5) 70 × 10^4^. **Legend:** (**A**–**D**) patients with hepatic cirrhosis with A child (**A**,**B**) or C child (**C**,**D**) stages by Child-Pugh score or hepatic steatosis (**E**,**F**) with acute intestinal inflammation (**A**,**C**,**E**) or chronic intestinal inflammation (**B**,**D**,**F**); (**G**) total reactive oxygen species count statistics. HC—hepatic cirrhosis; HS—hepatic steatosis; C—control; c—Child-Pugh score; CII; AII—chronic or acute intestinal inflammation.

**Figure 10 ijms-25-02472-f010:**
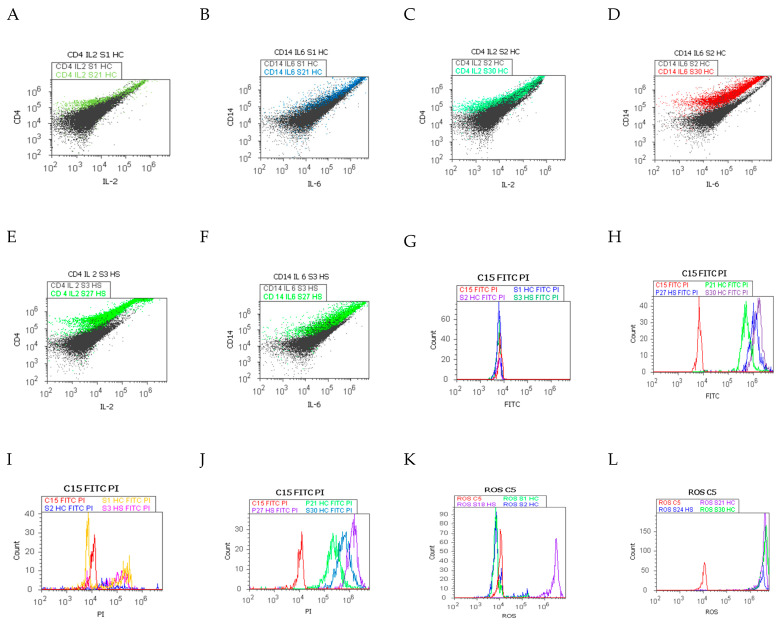
Dysregulated immunological barrier after endotoxins inflammatory action in gut cells by ROS-cell death mechanisms in the gut–liver axis. **Legend:** (**A**–**L**) experimental duodenal tissue samples recovered from patients with hepatic cirrhosis (HC) with A child (S1, S21) or C child stages (S2, S30) by Child-Pugh score or hepatic steatosis (HS) (S3, S27) with acute intestinal inflammation (S1, S2, S3, S18) or chronic intestinal inflammation (S21, S30, S27, and S24); (**A**–**F**) pro-inflammatory biomarkers patterns (CD4+/IL2+; CD14+/IL6+ dual stain by Alexa Fluor 488/PE); (**G**–**J**) cell death by Annexin V-FITC/propidium iodide (PI) dual stain; (**K**–**L**) oxidative stress by total reactive oxygen species count (ROS).

## Data Availability

Data are available in this manuscript.

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
