# Peer review of "Endotoxin Inflammatory Action on Cells by Dysregulated-Immunological-Barrier-Linked ROS-Apoptosis Mechanisms in Gut–Liver Axis"

_ijms, 2024, doi:10.3390/ijms25052472_

Round 1
Reviewer 1 Report
Comments and Suggestions for Authors
The paper entiteled “Endotoxin Inflammatory Action on Cells by Dysregulated Immunological Barrier Linked ROS- 2 Apoptosis Mechanisms in Gut-Liver Axis” presents the adapted changes in the inflammatory potential of the gut immune environment linked to oxidative stress-cell death mechanisms in chronic hepatic diseases.
Abbreviation section: Missing and I advise the authors to add it to their manuscript.
Figures: Too hard to read the figures. More attention should be given to the figures and present them in a clear way.
Minor editing of English language is required.
I request the authors to add a summary figure to summarize their main findings.
Comments on the Quality of English LanguageMinor editing of English language is required.
Author Response
Dear Reviewer 1,
Please see the attached document.
Best regards,
Ph.D. biologist Elena Matei
Corresponding author

Reviewer 2 Report
Comments and Suggestions for Authors
This study investigates the modified inflammatory profile of the gut immune environment associated with oxidative stress-induced cell death mechanisms in chronic hepatic diseases. The analysis focuses on gut immune barrier dysfunction using pro-inflammatory biomarkers, oxidative stress, and cell death assessments in duodenal tissue samples. Findings reveal a robust immune response in chronic inflammation associated with hepatic cirrhosis, emphasizing dysregulated gut immune barrier function. The study identifies distinctive reactive oxygen species-cell death mechanisms in both chronic and acute inflammation scenarios when gut cells are exposed to endotoxins, illustrating immune alterations in the gut-liver axis
I believe the introduction and discussion contain more information than necessary for understanding the paper.
Improve the quality of the figures. It is challenging to understand the images; the resolution is too low.
This paper presents a substantial amount of results. However, I wonder if all the graphs are necessary to be included in the body of the paper. Perhaps adding some of the arranged graphs would make the cytometry figures easier to understand (Figures 1, 3, 4, 6, 7, and 9).
I suggest the authors improve the methodology. Add the titration of the antibodies used instead of the volume.
Author Response
Dear Reviewer 2,
Please see the attached document.
Best regards,
Ph.D. biologist Elena Matei
Corresponding author

Reviewer 3 Report
Comments and Suggestions for Authors
The authors aim to study the chronic and acute impact of inflammation on the intestine and the impact on immune barrier dysfunction and eventually affecting the gut-liver axis.
In the abstract and the introduction as well, a clear aim and hypothesis of why this study was conducted was not clear. The presentation is super choppy and needs better flow of information.
My overall and only comment is that the English needs to be heavily revised for the readers to understand.
Comments on the Quality of English LanguageNeeds reworking.
Author Response
Dear Reviewer 3,
Please see the attached document.
Best regards,
Ph.D. biologist Elena Matei
Corresponding author

Reviewer 4 Report
Comments and Suggestions for Authors
There are comments as shown below. Basically, this manuscript is uncompleted form.
1. What is the novelty or significance of the present study? Dysfunction of intestinal barrier has been demonstrated as authors mentioned in introduction.
2. Experimental condition is not comprehensive at all. What was done to achieve the objective of your study? Each parameter shown as the result was not appropriate to evaluate intestinal barrier function. In addition, significant relationship between each result cannot be understood since there is no explanation about it. The manuscript is not accepted as scientific research paper. Hence, I cannot help saying that it is essential to rewrite everything from scratch.
3. Background of abstract is not wired. This is not background of the present study and the objective is ambiguous, that should be revised.
4. The sentence in line44 as described “The liver is the last barrier for microorganisms that enter the bloodstream due to affected gut mucosal.” should be revised since it is not easy to read.
5. Which countries or regions shows liver cirrhosis as 11th cause of death of as shown line64 to 65? It should be clearly described.
Comments on the Quality of English LanguageWhat the authors want to say is not easy to be understood. Checking grammer by native and elaboration of sentenses are necessary.
Author Response
Dear Reviewer 4,
Please see the attached document.
Best regards,
Ph.D. biologist Elena Matei
Corresponding author

Round 2
Reviewer 3 Report
Comments and Suggestions for Authors
The authors have improved the quality of the paper.
All of the citations are missing from the manuscript however.
Author Response
Dear Reviewer 3,
Please see the attached document.
Kind regards,
Ph.D. biologist Elena Matei
Corresponding author

Reviewer 4 Report
Comments and Suggestions for Authors
comments are in attached file.

English is maybe OK but reconstruction of story should be done.
Author Response
Dear Reviewer 4,
Please see the attached document.
Kind regards,
Ph.D. biologist Elena Matei
Corresponding author

Round 3
Reviewer 4 Report
Comments and Suggestions for Authors
Nothing has been revised.
Comments on the Quality of English LanguageMinor check would OK.